# The Ecology and Evolution of the Baker’s Yeast *Saccharomyces cerevisiae*

**DOI:** 10.3390/genes13020230

**Published:** 2022-01-26

**Authors:** Feng-Yan Bai, Da-Yong Han, Shou-Fu Duan, Qi-Ming Wang

**Affiliations:** 1State Key Laboratory of Mycology, Institute of Microbiology, Chinese Academy of Sciences, Chaoyang District, Beijing 100101, China; handy@im.ac.cn (D.-Y.H.); duanshoufu@126.com (S.-F.D.); 2College of Life Sciences, University of Chinese Academy of Sciences, Shijingshan District, Beijing 100049, China; 3School of Life Sciences, Institute of Life Sciences and Green Development, Hebei University, Baoding 071002, China; wangqm@hbu.edu.cn

**Keywords:** *Saccharomyces cerevisiae*, ecology, evolution, population genomics, phylogeography, yeast domestication

## Abstract

The baker’s yeast *Saccharomyces cerevisiae* has become a powerful model in ecology and evolutionary biology. A global effort on field survey and population genetics and genomics of *S. cerevisiae* in past decades has shown that the yeast distributes ubiquitously in nature with clearly structured populations. The global genetic diversity of *S. cerevisiae* is mainly contributed by strains from Far East Asia, and the ancient basal lineages of the species have been found only in China, supporting an ‘out-of-China’ origin hypothesis. The wild and domesticated populations are clearly separated in phylogeny and exhibit hallmark differences in sexuality, heterozygosity, gene copy number variation (CNV), horizontal gene transfer (HGT) and introgression events, and maltose utilization ability. The domesticated strains from different niches generally form distinct lineages and harbor lineage-specific CNVs, HGTs and introgressions, which contribute to their adaptations to specific fermentation environments. However, whether the domesticated lineages originated from a single, or multiple domestication events is still hotly debated and the mechanism causing the diversification of the wild lineages remains to be illuminated. Further worldwide investigations on both wild and domesticated *S. cerevisiae*, especially in Africa and West Asia, will be helpful for a better understanding of the natural and domestication histories and evolution of *S. cerevisiae*.

## 1. Introduction

The yeast *Saccharomyces cerevisiae* preferentially metabolizes sugar by anaerobic fermentation to produce ethanol and CO_2_, even when oxygen is available for aerobic respiration. This aerobic fermentative trait known as the Crabtree effect [1] is thought to be an adaptative invention, which endows the yeast with a strong ability to compete with other microbes in sugar-rich niches by fast sugar consumption and ethanol production [2]. Owning to this distinct property, *S. cerevisiae* has been used by humans worldwide for brewing and baking for thousands of years. Evidence for wine or wine-like beverage production dates from about 7000 BC in China [3,4], 6000 BC in Iran [5], 4000 to 3100 BC. in Mesopotamia [6], and 3150 BC in Egypt [7].

Early studies in microscopy, microbiology, enzymology, and biochemistry performed one and a half centuries ago by pioneering scientists, including Antonie van Leeuwenhoek, Joseph Gay-Lussac and Louis Pasteur [8,9,10], led to the discovery of *S. cerevisiae* as an agent of fermentation. The isolation of pure yeast cultures in 1888 by Emil Hansen, who perfected the method of Louis Pasteur [11], paved the way for extensive use of *S. cerevisiae* in biological research. Since then, *S. cerevisiae* has become one of the most powerful eukaryotic models in virtually every discipline of biology. In 1996, *S. cerevisiae* became the first eukaryote to have its genome completely sequenced [12]. The availability of the high-quality reference genome of *S. cerevisiae* together with a series of yeast strain libraries, including gene deletion libraries [13], has greatly facilitated the research of functional genomics and systems and synthetic biology, resulting in many remarkable developments in recent years, such as the artificial synthesis of yeast chromosomes [14] and the creation of a functional single-chromosome yeast [15].

However, basic research on yeast has been mainly based on laboratory strains, predominately S288C and its derivatives [16]. S288C is an artificially modified strain produced through numerous deliberate crosses with approximately 90% of its genome from strain EM93, which was isolated from a rotting fig collected in California’s Central Valley [16]. Population phenomic analysis of *S. cerevisiae* showed that strain S288C is highly atypical and represents a phenotypic extreme of the species [17], probably due to auxotrophic and other genetic markers in its genome. This highlights the limitation of inferring gene-trait connections in the species based on laboratory strains. Laboratory strains also provide very limited information about the ecology and natural history of the species. In recent years, an increasing number of wild lineages of *S. cerevisiae* with surprisingly high genetic diversity have been discovered from natural environments [18,19,20,21,22,23], stimulating the interest in understanding the natural history and function of the budding yeast in the wild. The discovery of wild lineages also provides a better framework for inferring the origin and evolution of domesticated populations of the yeast. In the past decades, thousands of wild and domesticated strains of *S. cerevisiae* have been characterized phenotypically and genomically [24], providing new insights into the ecology, environmental adaptation, population structure, biogeography, evolution, and natural and domestication histories of the species.

## 2. The Life Cycle of *S. cerevisiae*

The life cycle of *S. cerevisiae* has been well documented in the laboratory [25] (Figure 1). *S. cerevisiae* usually grows as a diploid in artificial nutrient-rich medium such as YPD (1% yeast extract, 2% peptone, and 2% dextrose) and reproduces clonally by budding, with an optimal growth temperature of around 30°C. It will sporulate and undergo meiosis in response to nitrogen starvation, resulting in the formation of four haploid spores in an ascus. Two of the spores have mating type a (*MAT*a) and the other two *MATα*. Mating type is determined by a single locus *MAT* in the middle of the right arm of chromosome III [26,27]. A pair of spores with opposite mating types can mate within the ascus upon germination (intratetrad mating or automixis) and form a diploid cell. Ascospores can also germinate to form haploid cells, which can reproduce mitotically by budding, resulting in the formation of *MAT*a and *MATα* haploid clones. However, the haploid phase usually only exists for a very short period in the life cycle. A haploid cell can mate with another haploid with an opposite mating type either from a different ascus of the same strain (selfing) or from a different strain (outcrossing or amphimixis). Haploid cells can also undergo a mating-type switch by exchanging types at the *MAT* locus via a gene conversion event (Figure 1). The molecular mechanism of mating-type switching is the replacement of the genetic factor of the *MAT* locus by a copy of the alternative factor located at a silent locus. There is one silent locus for each mating type (*HML* homologous to *MATα* and *HMR* homologous to *MAT*a). Recombination between *MATα* and *HMR*(a) or between *MAT*a and *HML*(*α*) mediated by the *HO* gene, which encodes an endonuclease that induces a double strand break of DNA within the *MAT* locus, results in a switch in mating type [27,28]. Mating-type switching occurs in an accurately regulated pattern that allows only half of the cells in a colony to switch mating types in any one cell division cycle and produces cells with opposite mating types in close proximity, thus facilitating cell mating to form diploid cells [27,28]. This process is termed haplo-selfing or autodiploidization (Figure 1). The similar mating-type switch mechanism involving the three-locus (*MAT*-*HML*-*HMR*) factors has been found in the majority of species in the *Saccharomycetaceae* clade studied [29,30]. Mating-type switch phenomena mediated by simpler ‘‘flip/flop’’ mechanisms have also been detected in at least 10 other groups of yeasts [30].

Different mating behaviors in *S. cerevisiae* have different genetic and evolutionary consequences [29,30,31,32]. Outcrossing by mating of haploids of different strains leads to the formation of a heterozygous diploid cell, while haplo-selfing results in the formation of an entirely homozygous diploid cell except for the mating type locus. Outcrossing rates of *S. cerevisiae* are estimated to be very low, at 9 × 10^−5^ to 2 × 10^−5^ per cell division [33,34]. However, outcrossing may be promoted by insect dispersal vectors such as *Drosophila* [35,36] and social wasps [37]. The high degree of homozygosity observed in wild strains of *S. cerevisiae* [20,21,38] implies the high frequency of haplo-selfing in nature. A “genome renewal” hypothesis has been proposed to explain the population genetic implications of haplo-selfing [39,40,41]. This hypothesis speculates that haplo-selfing makes deleterious recessive alleles homozygous and exposed directly to purifying selection, thus facilitating the purging of deleterious, and the fix of beneficial, alleles. However, such a proposed benefit is achieved at the cost of the loss of allelic variation, which might be of potential to meet other selective challenges [32].

The significance of the life cycle and breeding systems of *S. cerevisiae* characterized in the laboratory remains largely unknown in natural populations of the species. It is difficult to observe and directly characterize the growth profile, mating behavior and life cycle progress of the species in the wild. More ecological and population genetic studies on natural *S. cerevisiae* are required to examine the roles and consequences of sexual and asexual reproductions of the species in nature.

## 3. Habitats of *S. cerevisiae* in the Wild

For a long time, *S. cerevisiae* was considered an exclusively domesticated species because of its scarcity in natural environments [42,43]. Though *S. cerevisiae* was occasionally isolated from the wild, the feral strains were thought to be the escapees from domestic stocks [42,43,44,45]. However, an early field survey in Japan showed that *S. cerevisiae* was frequently isolated from forest materials including soil, decayed leaves, and tree bark [46], implying the common occurrence of the species in nature. Then, a growing number of studies also suggested that *S. cerevisiae* might be distributed in a wide range of forest habitats as well as vineyards [19,40,47,48,49,50,51].

The development of efficient selective isolation methods of *S. cerevisiae* from the wild has promoted field surveys of the species. *S. cerevisiae* strains can be readily isolated from alcoholic fermentation processes or other fermenting sugar-rich substrates using the standard dilution plating method, but can be hardly isolated from natural substrates using the conventional protocol due to complex microbial communities and low population density of the yeast in nature. Enrichment media containing ethanol was used to isolate *S. cerevisiae* from vineyard grapes [49,50]. An enrichment protocol with a medium consisting of 3 g yeast extract, 3 g malt extract, 5 g peptone, 10 g sucrose, 1 mg chloramphenicol, 1 mL of 1 M HCl and 76 mL ethanol per liter was successfully used for *S. cerevisiae* and *S. paradoxus* isolation from uncultivated habitats [19]. This ethanol enrichment method was then used in different field surveys of *Saccharomyces* species with minor modifications in carbon and nitrogen sources, and ethanol and antibiotic concentrations [20,52,53].

A systematic field survey of the distribution of *S. cerevisiae* in nature, with an unprecedented scale of diversified substrate sampling and climate zone coverage, was performed in China [20]. *S. cerevisiae* strains were successfully isolated from fruit, tree bark, soil, and rotten wood samples collected from orchards and cultivated, secondary, and primeval forests located in tropical to temperate climate zones, using an enrichment medium modified from Sniegowski et al. [19] containing 3 g yeast extract, 3 g malt extract, 5 g peptone, 10 g glucose, 2 mg chloramphenicol, 1 mL of 1 M HCl and 80 mL ethanol per liter. This field survey showed for the first time that *S. cerevisiae* is a ubiquitous species in nature, occurring in both man-made environments and habitats remote from human activity.

It is now clear that wild *S. cerevisiae* distributes ubiquitously in nature, but whether it is a nomadic microbe or prefers to live in specific niches in the wild is still uncertain [54]. Though *S. cerevisiae* was thought to be common in vineyards, it was rarely isolated from grape fruit [50,55]. Metagenomic sequencing also showed that the *Saccharomyces* species was vanishingly rare on ripe grapes in vineyards, compared with Crabtree-negative yeast species [56]. Based on the isolation result of *S. cerevisiae* from a total of 2064 samples, Wang et al. [20] showed that the success rate of *S. cerevisiae* isolation from fruit samples (6.5%, 59/753) was lower than those from rotten wood (9.2%, 39/425), soil (10.8%, 15/139) and tree bark (16.5%, 123/747) samples (Figure 2). *S. cerevisiae* was more frequently isolated from forest soil samples (success rate 13.7%, 7/51) than from orchard soil samples (9.1%, 8/88). Among the fruit samples harboring *S. cerevisiae*, the grape samples exhibited the lowest success rate of isolation (1.8%, 8/452) (Figure 2). These studies suggest that grape or other fruit is probably not the preferred niche of *S. cerevisiae* in the wild as predicted based on the observation that *S. cerevisiae* is dominant in sugar-rich fermentation environments.

*S. cerevisiae* and its sibling wild species *S. paradoxus* have been frequently isolated from oak tree bark [19,52,57,58], leading to the belief that oak is probably a preferred niche of *S. cerevisiae* in nature. However, Goddard and Greig [54] argued that this belief is likely based on the biased survey of many researchers who simply intended to isolate yeasts from wild environments where *Saccharomyces* has already been found. The large-scale survey performed by Wang et al. [20] showed that, among the different types of substrates sampled, coniferous tree bark samples (over 100) all showed negative *Saccharomyces* isolation, while broad-leaved tree bark exhibited a higher isolation rate of *S. cerevisiae* than those of other substrates as mentioned above. The 123 bark samples harboring *S. cerevisiae* represented more than 36 species of broad-leaved trees. Among the total 392 *Quercus* tree bark samples, 51 (13.0%) showed positive *S. cerevisiae* isolation. Interestingly, the bark samples from other Fagaceae trees, including *Castanea* (18.5%, 15/81), *Castanopsis* (57.9%, 11/19) and *Cyclobalanopsis* (55.6%, 5/9) showed higher *S. cerevisiae* isolation rates than from *Quercus* trees. The elm tree (*Ulmus macrocarpa*) bark samples also showed much higher isolation rates (28.6%, 10/35) than *Quercus* (Figure 2). The distribution of *S. cerevisiae* in Brazil, where native oaks or other Fagaceae species are absent, was investigated by Barbosa et al. [59]. The success rate for *S. cerevisiae* isolation from the bark of *Tapirira guianensis* (Anacardiaceae), a species common in Brazil, was 13%, while other tree bark samples collectively provided a 4% isolation frequency. The bark samples of *Quercus rubra*, an ornamental oak imported from North America, showed a success rate of up to 71%, but only seven samples were collected [59]. These studies imply that *S. cerevisiae* might prefer to live on Fagaceae as well as some other broadleaved trees in the wild. A more comprehensively designed systematic survey is required to reveal the ecological niches of *S. cerevisiae* in nature.

**Figure 2 genes-13-00230-f002:**
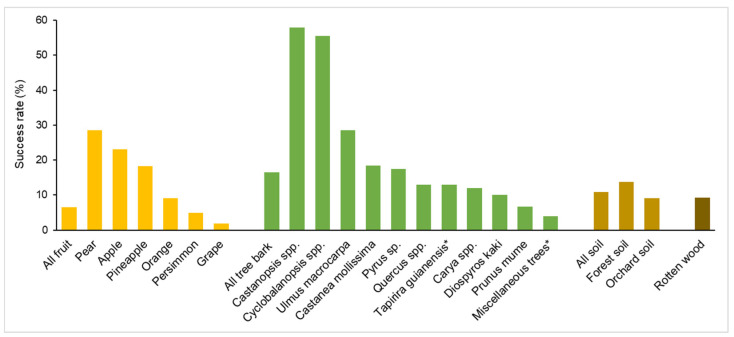
Success rates of *S. cerevisiae* isolation from different substrates in the wild. Data are from Wang et al. [20] except those marked with an asterisk which are from Barbosa et al. [59]. The substrates with more than ten samples subjected to *S. cerevisiae* isolation are selected. The types of the substrates (fruit, tree bark, soil and rotten wood) are distinguished using different colors and the specific substrates in each group are arranged according to the success rates of *S. cerevisiae* isolation.

## 4. Genetic Diversity and Population Structure of *S. cerevisiae*

The approximately 9000 year domestication history of yeast [3] is similar to that of key plants and animals, which usually have a domestication history of around 10,000 years [60]. The domestication of plants and animals has been extensively studied since Darwin [60,61], however, research centering on the domestication of yeast has rarely been performed until recently. The lag was partially due to the lack of reference wild populations of *S. cerevisiae* and poor understanding about the natural history of the yeast. The phylogenetic distinction between wild and domesticated populations of *S. cerevisiae* was shown for the first time in 2005 [62] based on sequence analysis of five genes (*CCA1*, *CYT1*, *MLS1*, *PDR10*, and *ZDS2*) and their promoters in 81 strains. Aa et al. [63] sequenced four loci (*CDC19*, *PHD1*, *FZF1* and *SSU1*) in 27 natural strains of *S. cerevisiae* and revealed a distinct population structure in the species. Strains from an oak forest and those from vineyards were clearly separated, probably due to ecological differences between the host trees. Since then, a growing number of population genetics and genomics studies have recognized a variety of phylogenetically distinct lineages of *S. cerevisiae* from natural and man-made environments. Liti et al. [64] performed a population genomics analysis of 38 *S. cerevisiae* strains with worldwide origins and recognized five main lineages, namely, Malaysian, North American, Sake, West African, and ‘Wine/European’, and many mosaics strains. This study also showed that the genome sequence diversity of the worldwide *S. cerevisiae* strains was very limited, equivalent to only a single *S. paradoxus* population. A similar population structure of *S. cerevisiae* consisting of the five lineages was revealed by the 100-genomes project [65], with the mosaic group being expanded to include clinical strains. Based on phylogenetic analysis of genome-scale single nucleotide polymorphisms (SNPs) extracted using microarray analysis, Schacherer et al. [66] recognized three distinct lineages (wine, sake and laboratory) from 63 *S. cerevisiae* strains. Cromie et al. [67] used a RAD-seq (restriction site-associated sequencing) strategy to genotype 262 *S. cerevisiae* strains and recognized geographically defined populations, namely, European, North American, Asian, and African/S. E. Asian, together with small groups from specific fermentation environments.

The *S. cerevisiae* strains employed in the early studies of population genetics and genomics were mainly from fermentation and human-associated environments, and wild strains were poorly represented. The wild strains designated in these studies were mainly from vineyards, oak tree bark and associated soil. Though the oak strains were considered “truly wild” in these studies, the association of the oak strains with human activities cannot be excluded, because the oak trees sampled were usually located in man-made environments or environments frequently visited by humans, such as parks or arboreta [19,63]. Wang et al. [20] revealed, for the first time, a surprisingly high genetic diversity of *S. cerevisiae* in the wild. They performed population genetics analysis based on the sequences of nine genes and four intergenic loci in 102 natural Chinese *S. cerevisiae* strains with different geographical and ecological origins, including over 30 strains from primeval forests. The wild strains exhibited a strong population structure consisting of highly diverged lineages without admixture. Eight new distinct wild lineages (CHN I–VIII) were identified from the Chinese strains. Most of the primeval forest strains occurred in ancient and significantly diverged basal lineages, while those from orchards and cultivated forests generally clustered in less differentiated domesticated or mosaic groups. The result suggests that greatly diverged wild populations of *S. cerevisiae* predate and exist independently of domesticated populations. This finding also dispels the suspicion of *S. cerevisiae* as a model in ecology and biogeography because of its close relationship with human activity and the potential complications of domestication [68].

The Chinese wild strains of *S. cerevisiae* characterized in Wang et al. [20] contribute the majority of the genetic variation of *S. cerevisiae*, with nearly double the combined sequence diversity of strains from the rest of the world documented by that date [64]. In addition, the basal lineages (CHN I–V) represent the oldest lineages of *S. cerevisiae* that have not been found outside China. These results provide strong evidence supporting the hypothesis that *S. cerevisiae* originated from Far East Asia [69].

In recent years, more *S. cerevisiae* strains with more diversified ecological and geographic origins have been sequenced by different research groups in the world. The genome sequence data of more than 2300 *S. cerevisiae* strains have been released publicly [22,24,70,71]. These strains are from a total of 93 countries or regions of the world, but the majority of the strains are from a limited number of countries (Figure 3A). The ecological origins of the strains are also biased, and wine, beer and clinical strains are over-represented (Figure 3B).

The largest genome sequence project of *S. cerevisiae* was performed by Peter et al. [72] who sequenced 918 strains using the Illumina platform and incorporated another 93 strains that had previously been sequenced. This collection of 1011 strains greatly expanded the breadth of the ecological and geographical origins of *S. cerevisiae*. Phylogenomic analysis of these strains identified 26 clades, including 10 domesticated and 11 wild clades and five clades without clear designation. The strains outside these distinct clades were included in three mosaic groups. Phylogenomic and principal component analyses based on genome wide SNPs support a single ‘out-of-China’ origin for *S. cerevisiae* [72].

*S. cerevisiae* strains from China, the center of origin of the species, are essential for illuminating the evolutionary history of the species. However, neither wild nor domesticated strains from China were sufficiently represented in the 1011 strains project, which contained only nine wild and two domesticated strains from mainland China. Duan et al. [21] sequenced the genomes of 106 wild and 160 domesticated *S. cerevisiae* strains from different sources in China, including the oldest wild lineages from primeval forests and domesticated lineages associated with ancient fermentation processes. In addition, a total of 287 *S. cerevisiae* strains with worldwide origins sequenced in previous studies were integrated in the phylogenomic analysis. The clear phylogenetic separation between the wild and domesticated populations was resolved. The wild strains from China were clustered into ten distinct lineages. In addition to the wild lineages recognized in Wang et al. [20] based on multilocus phylogenetic analysis, two novel lineages (CHN-IX and CHN-X) were identified (Figure 4). The discovery of CHN-IX, representing the oldest lineage of *S. cerevisiae*, resulted in an approximately one-third increase in the global genetic diversity of *S. cerevisiae* [21], reinforcing the Far East Asian origin hypothesis of the species.

The domesticated population of *S. cerevisiae* contains two major groups generally associated with liquid-(LSF) and solid-state fermentation (SSF) processes, respectively [21]. The SSF group contains the Baijiu (Chinese distilled liquors), Huangjiu (rice wines), Qingkejiu (highland barley wines), and Mantou (steamed bread) 1 to 7 lineages (Figure 4). The strains in the SSF group are exclusively from Far East Asia. The Sake lineage recognized in previous studies represents a sub-clade of the Huangjiu lineage involved in rice wine fermentation. The LSF group contains strains from both Asian and Western countries, including the ADY (active dry yeast), Milk/Cheese, Wine, Beer 1 and Beer 2 lineages and a Mixed lineage containing bread strains. The ADY lineage mainly consists of commercial yeast strains usually used for dough leavening (solid-state fermentation) in bread baking. This lineage is clustered in the LSF group closely related with the beer and wine lineages (Figure 4), suggesting that commercial bread strains were probably initially developed from beer or wine strains in Europe. The genetic diversity of domesticated strains from China is significantly higher than that from other regions of the world, implying that China/Far East Asia is also the origin center of the domesticated population of *S. cerevisiae* [21].

In previous population genomic studies of *S. cerevisiae*, African strains were poorly represented in terms of ecological and geographic origins, though Africa has a long history of fermented food production and diversified fermented foods [73,74]. Indeed, an African origin hypothesis for domesticated yeast has also been proposed [62]. Recently, Han et al. [22] sequenced 64 *S. cerevisiae* strains from indigenous fermented foods and forests in different African countries and performed an integrated phylogenomic analysis together with a collection of 486 isolates sequenced in previous studies [21,72,75]. These strains represented the maximum genetic diversity and almost all the recognized lineages of the species documented so far worldwide. The result confirmed the clear separation between the wild and domesticated populations of *S. cerevisiae* and the distinction between the LSF and SSF groups in the domesticated population (Figure 4). In addition to the African palm wine and West African Cocoa lineages recognized previously [67,72], five new lineages, namely, Mauritius/South Africa, South African beer, West African beer and African honey wine were recognized from the African strains (Figure 4). The African palm wine lineage is closely related to the Asian Islands lineage recognized in [72] in the wild population; the Mauritius/South Africa lineage was located basal to the SSF group, and the African Cocoa, beer and honey lineages were resolved as basal lineages of the LSF groups (Figure 4). The result suggests a potentially high genetic diversity and a long domestication history of *S. cerevisiae* in Africa [22].

Quantitative analysis based on a genome sequence data set from 612 *S. cerevisiae* strains representing the maximum genetic diversity and all known lineages of the species [22] showed that the sequence diversity (π) of all the strains was 6.51 × 10^−3^. The strains from China exhibited a moderately higher sequence diversity (π = 6.30 × 10^−3^) than the strains from rest of the world collectively (π = 5.95 × 10^−3^). The sequence diversity of the wild population (π = 8.08 × 10^−3^) was 1.48-fold higher than that of the domesticated population (π = 5.46 × 10^−3^) (Figure 5). The CHN-VI/VII and the Milk/Cheese lineages showed the highest intra-lineage sequence diversities among the wild and domesticated lineages, respectively; and the CHN-IX and West African Beer lineages exhibited the highest inter-lineage divergence (Figure 5).

## 5. Origin of the Domesticated Population of *S. cerevisiae*

Previous studies generally support the China/Far East Asia origin hypothesis of *S. cerevisiae*. Ancient basal lineages of *S. cerevisiae* have not been found outside China, despite extensive survey in Europe [52,76], North America [19,67], South America (including Amazonian rainforests) [59], New Zealand [55,77,78] and Africa [22,79,80]. However, the origin of the domesticated population of *S. cerevisiae* is still a debated issue [81]. Basically, two hypotheses have been proposed: (1) Chinese or Asian wild *S. cerevisiae* strains immigrated to other regions and were then domesticated independently in different areas [64,72]; or (2) after a single ancestral domestication event occurring most likely in China or Asia, domesticated ancestors were later introduced to other regions [21,22].

A population genomics study mainly on ale beer and wine yeasts showed that present industrial yeasts originated from only a limited number of ancestors [75], but the ancestors were not specified. Other studies [64,72] revealed close relationships of different domesticated lineages with different wild relatives of *S. cerevisiae*, suggesting that multiple independent domestication events led to the origin of various domesticated lineages. This multiple domestication events scenario was also supported by additional studies based on different strain and data sets [62,71,78,82,83]. However, the closest local wild relatives of individual domesticated lineages have not been specified, except for the wine lineage. An African and Mesopotamian origin hypotheses of wine strains were proposed by Fay and Benavides [62] and Legras et al. [82], respectively, but the numbers and geographic and ecological origins of the strains employed in these studies were quite limited, and the hypotheses were not supported in subsequent studies. Almeida et al. [76] showed that the Mediterranean oak (MO) lineage was the ancestor of the European Wine lineage. However, this hypothesis is not supported by recent studies employing more *S. cerevisiae* strains with more diversified geographic and ecological origins [21,22,71,72]. The close relationship of the MO lineage with the Wine lineage was not resolved in either of these studies.

On the other hand, the studies of Duan et al. [21] and Han et al. [22] showed the clear separation between the wild and domesticated populations of *S. cerevisiae*. The domesticated lineages recognized worldwide so far form a monophyletic clade sharing a common ancestor, suggesting a single domestication event scenario. The single domestication event hypothesis is also supported by the following observations. First, the genetic diversity of the domesticated population is significantly lower than that of the wild population, exhibiting a bottleneck as the domesticated population diverged from the wild population of *S. cerevisiae* [21,22]. Second, almost all of the domesticated strains are heterozygous and almost all the wild strains, especially those from primeval forests, are homozygous, implying that the domesticated strains share a heterozygous ancestor likely formed by outcrossing between genetically different wild strains [21]. Third, the domesticated lineages exhibit common expansion and contraction patterns of certain genes despite their different ecological origins. Interestingly, even though the Milk and Wine lineages in the LSF group are from niches without maltose, they harbor duplicated *MAL31*, *MAL32* and *MAL33* genes and share elevated maltose utilization ability with the other domesticated lineages from environments with maltose as a dominant carbon source [21]. Fourth, the domesticated lineages from Africa are located in the same monophyletic group as those from Asia, Europe and America [22]. The African honey wine strains also exhibit strong maltose utilization ability. These observations imply that the domesticated *S. cerevisiae* lineages likely originated from a heterozygous ancestor initially for adaptation to maltose-rich niches [21].

The failure to observe a clear separation between the domesticated and wild populations and to recognize the LSF and SSF groups of *S. cerevisiae* in other population genomic studies employing worldwide strains [71,72] were likely due to sampling bias as shown in Figure 3B. In these studies, domesticated strains associated with liquid-state fermentation were generally over-represented, while strains associated with solid-state fermentation and wild strains were quite limited. In the 1011 genome project, more than one third (35.8%) of the strains compared were located in the Wine/European clade [72]. The so-called wild strains employed in these studies were limited and usually from man-made environments such as vineyards and cultivated oak trees. A phylogenomics analysis based on a collection of *S. cerevisiae* strains with balanced strain numbers representing different groups or lineages, and the maximum global genetic and ecological diversity, is required to have a better understanding of the population structure of the species.

## 6. Intrinsically Different Life Strategies of the Wild and Domesticated Populations of *S. cerevisiae*

Previous studies have shown that *S. cerevisiae* occurs in both natural and man-made environments with high genetic diversity and clear population structure. However, different studies resulted in different answers to a fundamental question of whether the diversity of *S. cerevisiae* is primarily driven by niche adaptation and selection, or neutral genetic drift, echoing the long standing selectionist vs. neutralist debate in evolutionary biology. Some studies show that *S. cerevisiae* strains are principally organized by geography, highlighting the role of genetic drift in shaping the population structure of *S. cerevisiae* [64,67,78,83], while others recognize mainly ecologically defined populations, suggesting that natural selection may play a more important role than geographic factors in the diversification of *S. cerevisiae* [62,63,66,84,85]. Recent studies suggest that the forces driving the evolution of *S. cerevisiae* are more complicated, and neither geographic nor ecologic factors can fully explain the population structure of the species. Different levels of divergence and different lineages may have resulted from different driving forces.

In general, ecology seems to be the primary force driving the evolution of *S. cerevisiae*, since the wild and domesticated populations are distinct phylogenetically and the domesticated population is apparently an outcome of natural and artificial selection for adaptation to nutrient- or sugar-rich environments [21,22,62]. Extensive adaptive genome variations, including different patterns in heterozygosity, SNPs, gene contents and copy numbers, and allele distributions have been observed between wild and domesticated populations [21,22,72], suggesting that wild and domesticated populations have evolved different life strategies for adaptation to generally different environments.

As mentioned above, field surveys have shown that *S. cerevisiae* is rare on fruits but frequently found from broad-leaved tree barks in the wild, implying that the wild strains are frequently subjected to various stresses, including starvation, aridity and cold, especially in temperate regions and winter. Duan et al. [21] showed that *FLO* genes, which are required for adhesion and biofilm formation in yeasts [86], are generally contracted or lost in the domesticated population but are maintained in the wild population, suggesting that cell adherence and biofilm formation are important for the yeast to cope with severe stresses in the wild. Another possible strategy of the yeast to survive in nature is through the formation of asci and ascospores. Duan et al. [21] showed that the sporulation rate of the wild population is significantly higher than that of the domesticated population. The majority of the wild strains sporulate well, while most of the domesticated strains fail to sporulate. Furthermore, more than 95% of the ascospores formed by the wild strains are viable but only less than 20% of the ascospores formed by the domesticated strains are viable. Sporulation is believed to be a developmental response of *S. cerevisiae* to nutrient limitation [87]. Ascospores are usually thick walled and contain concentrated cytoplasm and thus are more stress tolerant than vegetative cells [36,88]. Furthermore, the asci of *S. cerevisiae* are persistent with a rigid ascus membrane [89], providing additional protection to ascospores. Wild *S. cerevisiae* cells likely exist in a non-dividing quiescent state in much of their time [90]. Efficient sporulation might be selected for *S. cerevisiae* in the wild favoring a “hunker down” strategy [91]. A negative correlation of heterozygosity with sporulation and spore viability rates has been observed in *S. cerevisiae* [21,39,41]. Thus, high level homozygosity is likely required for efficient sexuality and sporulation of wild *S. cerevisiae* strains.

On the other hand, asexual reproduction and heterozygosity are likely adaptive traits of the domesticated population of *S. cerevisiae*. Domesticated strains usually live in nutrient-rich environments together with many other microbes and thus rapid cell proliferation is apparently profitable for the competition of *S. cerevisiae* with other microorganisms. Sexual reproduction is more costly than asexual reproduction and thus the latter was likely selected in domesticated strains. The stresses encountered by yeast cells in fermentation environments include high osmolarity induced by sugar substrates, elevated temperature and ethanol concentration, and reactive oxygen species (ROS) derived from oxygen metabolism [92]. Heterozygous yeast cells are usually more vigorous than homozygous cells in growth and stress tolerance because of heterosis, or hybrid vigor [41,93,94,95,96,97]. Recently, Song et al. [98] showed that thermotolerant heterosis is prevalent in the F1 hybrids formed by spore-to-spore mating of different wild *S. cerevisiae* strains. The hybrids can usually cope with oxidative stress more effectively by up-regulation of their one-carbon metabolism and related pathways, leading to reduced DNA and protein damage and higher energy use efficiency. These studies explain the prevalence of heterozygosity in the domesticated strains.

Although the *FLO* genes required for cell adhesion are beneficial for the survival of wild lineages of *S. cerevisiae* as mentioned above, in nutrition rich niches, cell adhesion may not be a required trait. In contrast, planktonic cells may have an advantage of rapid cell proliferation in fermentation environments. Thus, the *FLO* genes are generally lost or contracted in the domesticated lineages of *S. cerevisiae* [21]. The strong flocculation ability of some commercial strains for wine, beer and bio-ethanol production [99] is apparently a post-domestication trait resulting from artificial selection for yeast cell separation from final products after liquid-state fermentation. Elaborately bred pure yeast cultures are usually used in these fermentation processes.

## 7. The Diversity of the Domesticated *S. cerevisiae* Is Primarily Driven by Ecology

Ecology apparently plays a main role in the divergence of the domesticated lineages of *S. cerevisiae*. Osmolarity seems to be the primary selection pressure, since strains associated with liquid- and solid-state fermentation are clearly separated [21,22]. The main difference between the two types of fermentation is the water content of the substrates. The water contents are usually 80–90% and 40–60% in the liquid- and solid-state fermentation, respectively [22]. Within each of the LSF and SSF groups, strains associated with different food and beverage fermentation usually form distinct lineages. Remarkably, strains for the fermentation of grape juice, wort, milk, agave juice and honey, cluster in the Wine, Beer, Milk/Cheese, Mexican Agave and African Honey Wine lineages in the LSF group, respectively; while strains for the fermentation of dough, sorghum grain, barley grain, and cooked rice form the Mantou, Baijiu, Qingkejiu, and Huangjiu/Sake lineages in the SSF groups, respectively, regardless of their geographic origins (Figure 4) [21,22,72,100].

Extensive genetic variations leading to consequent phenotypic trait changes for adaptation to specific niches have been identified in different domesticated lineages [21,22,24,101,102]. Three unique HGT fragments (regions A–C) from *Zygosaccharomyces bailii* were identified from wine yeast strains [103]. These regions harbor key functional genes in wine fermentation and thus are believed to contribute to the adaptation of wine yeast strains to grape juice fermentation. Genes in these regions have also been found from other lineages, but are mostly limited in the LSF group [21,22]. The Alpechin, Brazilian bioethanol, Mexican agave and French Guianan lineages in the LSF group possess abundant introgressions from *S. paradoxus*. The Alpechin lineage carries the largest amount of *S. paradoxus* introgressions [72,104], implying the contribution of these introgressed material to the adaptation of the Alpechin lineage which occurs in olive oil related niches [105]. Another interesting example is the Milk/Cheese lineage of *S. cerevisiae*. Though *S. cerevisiae* is unable to utilize lactose, it is one of the dominant species in spontaneously fermented milk products containing lactose as the sole carbon source [106,107]. The milk-adapted yeast lineage has autonomously swapped all its structural *GAL* genes (*GAL2* and the *GAL7-GAL10-GAL1* cluster) with early diverged versions through introgression [21,100,108]. The rewired *GAL* network expresses constitutively and circumvents glucose repression through synergetic changes in the regulatory components of the network, resulting in a galactose-over-glucose preference switch and galactose-utilization rate elevation [108]. These changes enable the yeast to use galactose first, as soon as it is released from lactose hydrolyzation by co-existing lactose-fermenting microbes, and to minimize carbon source competition with other microbes which usually prefer glucose. Furthermore, the introgressed *GAL2* has been duplicated, enabling the yeast to transport both galactose and glucose faster, and concurrently utilize the two sugars. The adaptive ‘reverse’ evolution of the *GAL* network offers a competitive advantage to the milk/cheese yeast living in fermented dairy products together with other microbes [108].

Loss of heterozygosity (LOH), which enables the expression of recessive alleles and generation of new allele combinations, may play a role in niche adaption of diverse strains of *S. cerevisiae* [72,91,109,110,111]. Large scale whole-genome sequence analysis shows that LOH events are common in different domesticated lineages of *S. cerevisiae* [72]. Experimental evidence has shown that LOH contributes to the emergence of resistant mutants to the antifungal drug nystatin in laboratory populations [109], to the adaptation to nutrient-limiting conditions in hybrid yeast [110], or to rapid evolution of diploid genotypes under divergent selection [111]. However, lineage-specific LOH patterns have not been reported; therefore, the role of LOH in the adaptation of individual lineages to specific niches is still unclear.

Geography may also play a role in the divergence of domesticated lineages. The strains involved in Mantou (steamed bread) fermentation from different areas of China form several separate lineages (Mantou 1–7) [21,22]. The British, U.S. and Belgium/German beer strains are clustered in different subclades [75] and the South and West African beer strains form separate lineages [22] (Figure 4). Since the raw materials and fermentation processes may be different in different regions or countries, the role of ecological factors for the divergence of these domesticated strains cannot be excluded.

## 8. The Diversification of the Wild *S. cerevisiae* Is Largely Consistent with a Neutral Model

The genetic diversity of the whole species *S. cerevisiae* is mainly contributed by its wild population (Figure 5), which is clearly structured with highly diverged lineages [20,21,22,23]. Broadly, geography seems to play a main role in the diversification of the wild strains. Strains from forests in different countries or regions usually form different lineages, such as the North American Oak, Far East Russia, Ecuador, and Malaysian lineages (Figure 4) [22,72]. The *S. cerevisiae* strains from Amazon forests in Brazil also formed different lineages [59]. Within China, the primeval forest strains from south China are generally not mixed with those from north China [20,21,22]. The forest strains from different regions (Shaanxi and Beijing) in north China also form different lineages (CHN-II and CHN-IV, respectively). However, the role of ecological factors cannot be excluded because different countries and regions may be ecologically different. The flora in tropical and subtropical forests in southern China are different from those in the temperate forests in northern China [112].

Conversely, the high genetic diversities of wild strains from single locations have been well documented. Primeval forest strains from a single location may belong to highly diverged lineages, exhibiting a sympatric differentiation phenomenon [20,21,22]. An interesting example is the *S. cerevisiae* population in the tropical island Hainan, which is located in southern China with an area similar to Belgium in size. The Hainan strains characterized were from rainforests located in the southern part of the island with similar flora, but distributed in three highly diverged basal lineages CHN I, III and V [20,21,22]. Wild strains from the same habitat (rotten wood) collected in the same mountain (Wuzhi Mountain) also clustered in different lineages (CHN III and V). The wild strains from tree bark collected in a single subtropical primeval forest in Hubei, central China, were separated into two distantly related lineages CHN IX and X [21,22]. Therefore, ecology which is usually considered as the possible cause of sympatric differentiation, seems unable to explain the sympatric coexistence of the highly diverged lineages of *S. cerevisiae*. Indeed, positive and purifying selection was rarely detected in the Chinese wild population, supporting a neutral model for the evolution of the wild *S. cerevisiae* [21].

Long distance migration of wild *S. cerevisiae* strains seems to occur frequently in nature. In addition to human activity, animal vectors (e.g., insects and birds) may play a role in the immigration of *S. cerevisiae* in nature [36,78]. Insects may have a mutualistic relationship with *S. cerevisiae* [37]. Yeast spores are not adapted for wind-borne transmission like bacterial and other fungal spores, but can be carried on grapes or other fruit via insects [36,113]. Indeed, a single wild lineage may contain strains from geographically well separated regions. For example, among the basal wild lineages, the CHN-IX lineage formed by strains from Hubei, central China recognized in Duan et al. [21] and the Taiwanese lineage recognized in Peter et al. [72] actually belong to a single lineage [22,114]. Each of the CHN-I, V and X lineages contains strains from two to four different provinces in China; the CHN-IV lineage contains strains from different countries, including China, Japan and Russia; and the Ecuador/USA lineage contains strains from South and North America (Figure 4) [22]. These observations suggest that immigration and secondary contact of wild *S. cerevisiae* strains from different lineages is frequent in nature. However, genetic admixture between different lineages has rarely been detected in the wild population of *S. cerevisiae* [20,21,22], suggesting that reproductive isolation between different wild lineages is well established.

Previous studies show that large-scale chromosomal rearrangements might play a role in the onset of reproductive isolation in *S. cerevisiae* [20,115,116]. However, spore viabilities of crosses between strains from different wild lineages ranged from 10.2% to 89.1% [20], being much higher than those (usually less than 1%) of the crosses between different species of the genus *Saccharomyces* [115]. The partial or weak reproductive isolation is unable to explain the significant divergence of the wild lineages of *S. cerevisiae* and the lack of admixture among the wild lineages even though secondary contact frequently occurs due to human and animal activities.

## 9. A Modified Genome Renewal Hypothesis for Explaining the Diversification of *S. Cerevisiae* in the Wild

The life cycle and mating behaviors of *S. cerevisiae* (Figure 1) probably contribute to the reproductive isolation and diversification of wild strains. As discussed above, efficient sporulation might be a selected trait for *S. cerevisiae* to survive in the wild [32]. Repetitive starvation and aridity pressures in the wild would select for the capability to return efficiently to a diploid state, which is necessary for sporulation. Autodiploidization mediated by mating-type switch and intratetrad mating would apparently provide a selective advantage because these processes avoid the risk of the absence of adjacent mates with opposite mating types [29,32]. Multiple reinventions of mating-type switching have occurred during the evolution of budding yeasts, suggesting strong natural selection in favor of this property [30]. The seemly obligate homothallism of the wild *S. cerevisiae* probably prevents outbreeding and genetic admixture. On the other hand, mutation or occasional outbreeding due to population admixture of the wild *S. cerevisiae* caused by human or animal (insect) activities could create heterozygous strains. Reinstatement to a homozygous state of heterozygous strains due to haplo-selfing would produce new genotypes as predicted by Mortimer’s genome renewal hypothesis [39,40,41]. In the case of wild strain diversification, the hypothesis needs to be modified, because purging of deleterious alleles is not a necessary function of this model. The neutral polymorphisms due to mutation or outbreeding in the occasionally formed heterozygous strains in nature can be fixed via subsequent haplo-selfing, as illustrated in Figure 6. The modified genome renewal model can explain sympatric diversification observed in wild *S. cerevisiae*, for neither geographic nor ecological isolation is required in this model.

## 10. Conclusions and Future Perspectives

A global effort centering on the natural and domestication histories and evolution of *S. cerevisiae* in the past decades has revealed that *S. cerevisiae* distributes ubiquitously in nature, probably preferring to live on broad-leaved trees in the wild. The genetic diversity of the species is mainly contributed by its wild population which is clearly structured with highly diverged lineages. The genetic diversity of *S. cerevisiae* in China is significantly higher than in other regions of the world and the ancient basal lineages of the species have been found only in China, supporting an ‘out-of-China’ origin hypothesis of the species. Ecology seems to be the main force shaping the population structure of the species, resulting in the clear phylogenetic separation between the wild and domesticated populations, and the divergence of domesticated lineages associated with fermentations of different foods; while the diversification of wild strains seems consistent with a neutral model. The wild and domesticated populations exhibit hallmark differences in heterozygosity, sporulation rate and spore viability, suggesting intrinsically different life strategies of the two populations for adaptation to their generally different environments. Consistent CNVs, gene content and allele distribution variations between the wild and domesticated populations, and lineage-specific CNVs and HGT and introgression events leading to adaptation to specific niches, have been observed.

However, several outstanding questions concerning the evolution of *S. cerevisiae* remain to be addressed. The origin of the domesticated population of *S. cerevisiae* is still uncertain. There is an ongoing debate on whether different domesticated lineages originated independently in different places from different wild ancestors immigrated from Asia, or from a single ancestral domestication event occurring most likely in Asia which then diverged due to nature and artificial selection. Further worldwide investigations on both wild and domesticated *S. cerevisiae* will certainly be helpful to resolve the problem. The geographic and ecological origins of the sequenced *S. cerevisiae* strains are quite biased (Figure 3) and the Wine/Europe clade is much more heavily represented than the other clades (Figure 3B). A recent study shows potentially high genetic diversity and long domestication history of *S. cerevisiae* in Africa [22]. However, the genetic diversity of both the wild and the domesticated *S. cerevisiae* in Africa has not been fully investigated. It is unclear if primeval forests in Africa harbor any basal wild lineages of *S. cerevisiae*. West Asia also has a long history of fermented food production and a rich variety of fermented foods [117]. Some believe that wine fermentation technology may have originated in Mesopotamia and the Caucasus in West Asia before 6000 BC [118,119]. However, the West Asian population of *S. cerevisiae* is also very poorly represented in previous studies (Figure 3). If wild and domesticated strains from Africa and West Asia are respectively clustered in the same wild and domesticated groups together with those from East Asia, America and Europe, then the single domestication event will be supported. Conversely, if the model of multiple domestication events holds true, closely related wild relatives of local domesticated lineages will be found from the same continents or regions. If the single domestication event hypothesis is proved and if the original domestication event occurred in China, the wild and domesticated populations found outside China should originate respectively from the wild and domesticated ancestors from China. Then the ‘single’ out-of-China origin scenario [72] should be reconsidered.

The mechanism underlying the diversification of the wild *S. cerevisiae* population, especially the seemingly sympatric differentiation of the species in nature, needs to be revealed further. The combination of the observed prevalence of inbreeding in the wild *S. cerevisiae* strains, the life cycle and mating system of the species (Figure 1), and the modified Mortimer’s genome renewal model (Figure 6) can be used to explain the diversification of the wild *S. cerevisiae*. However, little is known about the life cycle of the species in nature. Direct in situ observation of wild strains and tracking and characterizing fluorescent labeled or bar-coded strains released in a controlled niche mimicking the wild environment could provide new insights into the life cycle progress and diversification of *S. cerevisiae* in the wild.

A hallmark domestication signature of *S. cerevisiae* is the strong ability to utilize maltose. Wild strains are usually unable to, or only very weakly, utilize maltose, while all domesticated strains tested, even though from niches (e.g., wine, honey, and milk) without maltose, can strongly utilize maltose [21]. Duplication of the *MAL* genes, especially the maltose transporter genes, has been observed in domesticated strains [21,120], explaining in part their elevated maltose utilization ability. However, it is unclear why wild strains are unable to utilize this sugar, since all genes responsible for maltose metabolism have been found in their genomes [21]. The *MAL* genes usually have similar sequences with duplicated copies and concentrate in subtelomeric regions [121]; high quality genome assemblies based on long-reads sequence strategies will be required to illustrate the fine structure of the *MAL* pathway and the evolution of the pathway from the wild to the domesticated population of *S. cerevisiae*.

The abundant HGT and introgression events found in the genomes of diverse *S. cerevisiae* strains imply frequent gene flow between the yeast and other microbes in different niches. Indeed, *S. cerevisiae* usually coexists with other microbes in the wild and spontaneous fermentation environments. Mutualistic interactions of *S. cerevisiae* with bacteria have been reported in different studies [108,122,123,124,125]. The interaction between *S. cerevisiae* and lactose-utilizing microbes in fermented dairy products apparently contributes to the divergence of the Milk/Cheese lineage of *S. cerevisiae* [108]. The observed lineage-specific introgression/HGT events in different groups of *S. cerevisiae* [21] imply the role of these events to the differentiation and adaptation of these lineages. Further studies on the interaction of *S. cerevisiae* with other microbes in different habitats are required to understand the role, significance and mechanisms of microbial interactions in the evolution and domestication of *S. cerevisiae*.

Pangenome analysis based on 1011 strains recognized 4940 core open reading frames (ORFs) that are shared by all strains compared, and 2856 ORFs that are variable in different strains or lineages [72]. Principal component analysis (PCA) of gene contents recapitulated the differentiation of the main Wild, LSF and SSF groups that were recognized based on genome-wide SNPs [22]. These results suggest that variable genes are important for environmental adaption of different groups. Specific variable genes responsible for, or contributing to, the adaptation of different groups or lineages to different niches remain to be fully revealed.

*S. cerevisiae* is one of the Crabtree-positive and ethanol fermentation species in the genus *Saccharomyces* [126]. It is physiologically similar to its sibling species *S. paradoxus* [127] and usually co-exists with the latter in nature [128]. However, it is puzzling that only *S. cerevisiae* is domesticated. The wild species of *Saccharomyces* are usually homozygous, being similar to the wild population of *S. cerevisiae*. The domesticated *S. cerevisiae* strains are exclusively heterozygous, suggesting that heterozygosity is crucial for the fitness of *Saccharomyces* yeasts in nutrient-rich fermentation environments. Tolerance to high osmolarity, temperature, ethanol concentration, and ROS level is required for domesticated strains [92]. A recent study showed that heterosis contributes to high-temperature tolerance of domesticated *S. cerevisiae* [98], while another study showed that artificial hybrids of *S. paradoxus* did not show heterosis [129], implying heterozygosity may not contribute to the fitness of this species. Indeed, heterozygous strains of *S. paradoxus* and other non-*cerevisiae Saccharomyces* species have rarely been found in nature or man-made environments, though inter-species hybrids of *Saccharomyces* species are not uncommon in industrial yeast strains [101,128,130]. It will be interesting to investigate the ability to form intra-specific hybrids of non-*cerevisiae Saccharomyces* species and the fitness of these hybrids (if obtained) relative to their parents. These studies will, in turn, be helpful for a better understanding of the natural and domestication histories of *S. cerevisiae*.

## Figures and Tables

**Figure 1 genes-13-00230-f001:**
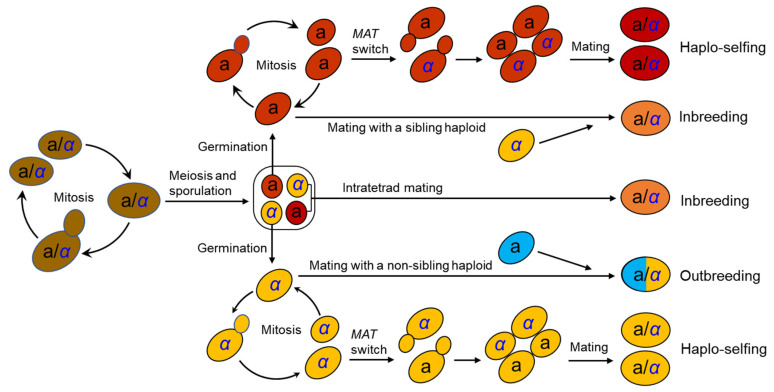
The life cycle and mating behaviors of *S. cerevisiae*. Vegetative cells are usually diploid and reproduce asexually by budding (mitosis). A diploid cell undergoes meiosis and sporulation due to nitrogen starvation and results in the formation of a tetrad with four ascospores, which either undergo intratetrad mating to form a diploid cell, or germinate to form haploid cells. A haploid cell either reproduces by budding, or mates with a sibling or non-sibling haploid with an opposite mating type to form a diploid cell, or undergoes haplo-selfing or autodiploidization through a process known as mating-type (*MAT*) switch to restore the diploid phase.

**Figure 3 genes-13-00230-f003:**
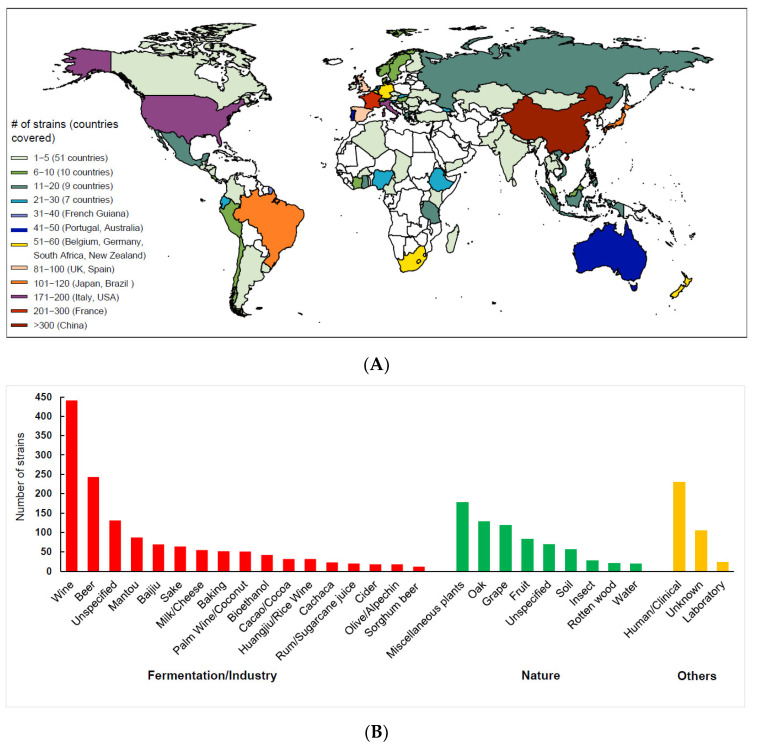
Geographic (**A**) and ecological (**B**) origins of the *S. cerevisiae* strains with their genome sequences available to the public.

**Figure 4 genes-13-00230-f004:**
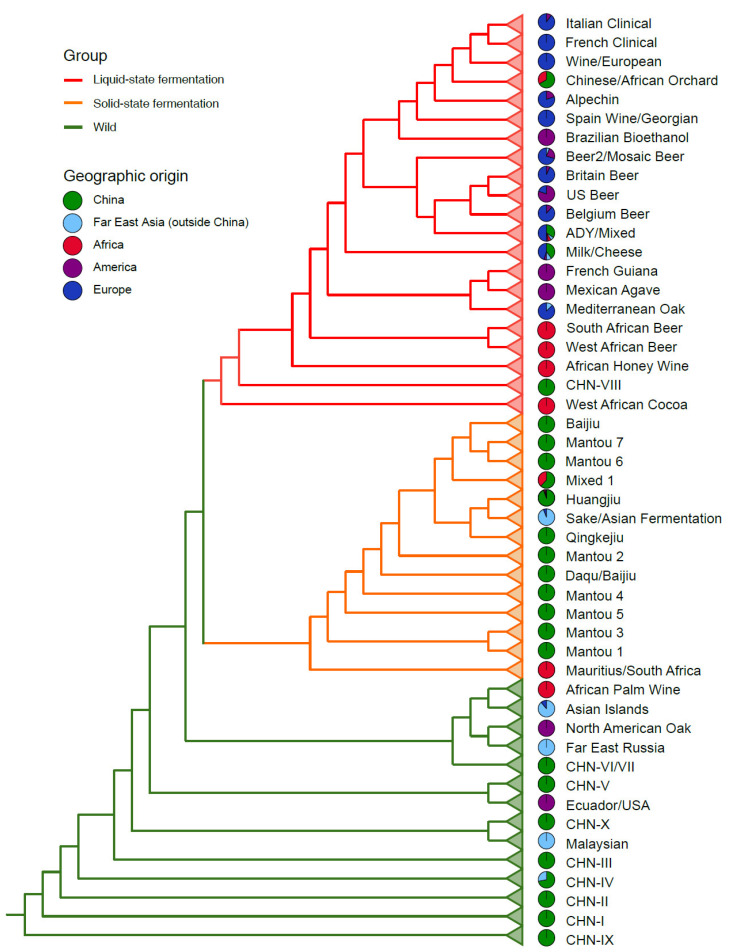
A schematic diagram showing the phylogenetic relationships of the recognized lineages of *S. cerevisiae*. The dendrogram is drawn according to the phylogenomic analysis performed by Han et al. [22] based on genome-wide SNPs from a set of *S. cerevisiae* strains representing the maximum global genetic diversity and almost all recognized lineages of *S. cerevisiae*, however, branch lengths do not exactly correspond to genetic distances between different lineages. The mosaic strains are not included. The wild and domesticated (liquid- and solid-state fermentation) groups are distinguished using branch lines with different colors. The pie charts represent the geographic origins of the strains in each lineage.

**Figure 5 genes-13-00230-f005:**
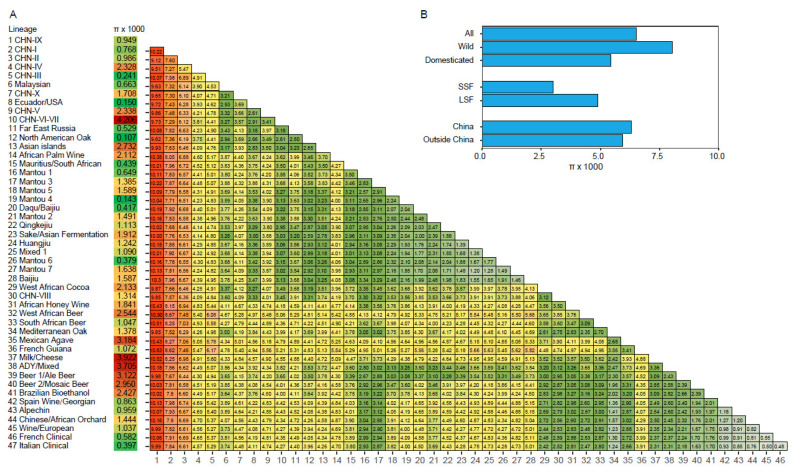
Sequence diversity within and divergence between different lineages (**A**), and sequence diversity of different groups (**B**), of *S. cerevisiae*. The data were calculated from the genome-wide SNPs from a set of 612 strains compared in Han et al. [22] representing the maximum global genetic diversity and almost all recognized lineages of *S. cerevisiae*. SSF, solid-state fermentation; LSF, liquid-state fermentation.

**Figure 6 genes-13-00230-f006:**
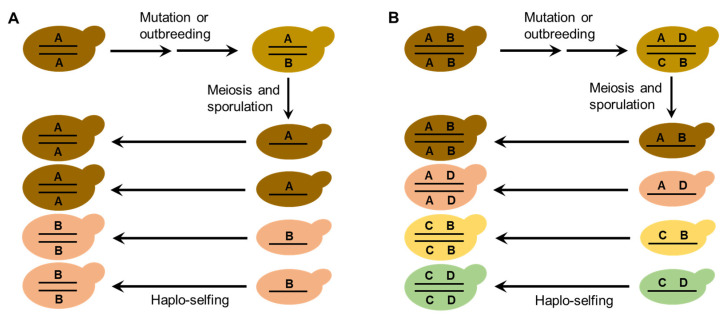
A schematic illustration of a modified Mortimer’s genome renewal model for explaining the diversification of wild *S. cerevisiae* strains. Theoretically, in the case of one neutral mutation in one locus (**A**), one new homozygous diploid cell line with a new genotype can be created; while in the case of two loci harboring one neutral mutation each (**B**), three new homozygous diploid cell lines with different genotypes can be created via meiotic recombination and haplo-selfing processes.

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
