# Peer review of "The Ecology and Evolution of the Baker’s Yeast *Saccharomyces cerevisiae"

_genes, 2022, doi:10.3390/genes13020230_

Round 1
Reviewer 1 Report
The authors present an interesting and well written review article about the possible origin and diversity of the yeast Saccharomyces cerevisiae, a major actor of natural and artificial fermentations as well as a major experimental model for biological research. The article is well organized and offers a significant list of references in a domain that have recently experienced increasing interest. I have only minor remarks that, if answered properly, would further increase the quality of the article.
paragraph 2 line 87. the mating type switching of S. cerevisiae does not allow a haploid cell to mate with its daughter because the switch occurs prior to the S phase and, threfore, both mother and daughter cells switch simultaneously. Daughter cell should be replaced by grand daughter (same for figure 1).
paragraph 2. the authors only discuss switching of S. cerevisiae without any reference to the switching in other Saccharomyciotina species. This is a pity given the recent progress in this field. And the fact that they correctly cite ref. 32.
paragraph 4. line 194. It would be desirable to briefly explain in what sense the domestication of yeast is similar to those of plants and animals.
lines 281 and 289: grammar. please correct "is consisted" by appropriate terms.
lines 322-324: the argument is excessive. 6.3 is only moderately higher than 5.95. This is explicitly visible in figure 5B. The major difference concerns wild versus domesticated, not china versus outside.
paragraph 5; lines 367-368. The argument is interesting but the sentence (almost all) is excessive. If there is a bias in homozygosity between wild and domesticated strains, it is only a trend, not an almost absolute property.
paragraph 6: the arguments are interesting but the authors should not forget that, if domesticated starins are more often geterozygous than wild ones, it suggest that they not only more often originate from outbreeding but also include long distance introgressions. Unfortunately, long distance introgressions are not properly discussed in this review despite that fact that some of them have been shown to play an important role in the domestication. Refrences to such articles would be appropriate in paragraph 7. Similarly, loss of heterozygosity (LOH) is not mentionned in paragraphs 8 and 9 despite the fact that it is a major phenomenon in yeast strains (see for exemple the works of Magwene et al or Peter et al ).
paragraph 10. the conclusion may be expanded given above remarks.
line 626: unique. Are the life cycle and mating system of S. cerevisiae so unique ? Actually, not if the authors would have paid little more attention on other Saccharomycotina species. Must be corrected.
Author Response
Response to Reviewer 1 Comments
The authors present an interesting and well written review article about the possible origin and diversity of the yeast Saccharomyces cerevisiae, a major actor of natural and artificial fermentations as well as a major experimental model for biological research. The article is well organized and offers a significant list of references in a domain that have recently experienced increasing interest. I have only minor remarks that, if answered properly, would further increase the quality of the article.
>> We highly appreciate these positive comments on our manuscript. We have revised the manuscript to accommodate every specific comment or suggestion from Reviewer 1.
paragraph 2 line 87. the mating type switching of S. cerevisiae does not allow a haploid cell to mate with its daughter because the switch occurs prior to the S phase and, threfore, both mother and daughter cells switch simultaneously. Daughter cell should be replaced by grand daughter (same for figure 1).
>> Thanks for pointing out this bug in our manuscript. We have revised in the text concerned (part 2, 1st paragraph, lines 96-100) and Figure 1.
paragraph 2. the authors only discuss switching of S. cerevisiae without any reference to the switching in other Saccharomyciotina species. This is a pity given the recent progress in this field. And the fact that they correctly cite ref. 32.
>> We have mentioned the similar mating-type switch in the majority of the species in the Saccharomycetaceae clade studied (part 2, 1st paragraph, lines 100-9104).
paragraph 4. line 194. It would be desirable to briefly explain in what sense the domestication of yeast is similar to those of plants and animals.
>> Here we mean that the lengths of the domestication histories of yeast and plants and animals are similar (largely around 10,000 years). We have revised the sentence (the 1st sentence in paragraph 1, part 4).
lines 281 and 289: grammar. please correct "is consisted" by appropriate terms.
>> We mean ‘contains’.
lines 322-324: the argument is excessive. 6.3 is only moderately higher than 5.95. This is explicitly visible in figure 5B. The major difference concerns wild versus domesticated, not china versus outside.
>> Here we intend to compare the diversity of S. cerevisiae from only one country with that from rest of the whole word. We think this means something. We have added the word ‘moderately’ to the sentence.
paragraph 5; lines 367-368. The argument is interesting but the sentence (almost all) is excessive. If there is a bias in homozygosity between wild and domesticated strains, it is only a trend, not an almost absolute property.
>> The argument is based on the study of Duan et al. (2018) showing the hallmark difference in heterozygosity between the wild and domesticated populations. The so called heterozygous wild strains reported in other studies are usually from orchards or cultivated forests belonging to man-made environments, not truly wild strains. The wild strains from primeval forests are exclusively homozygous, being similar to the wild species S. paradoxus. Homozygous strains have not been detected in our studies and have rarely been reported in other studies. Nevertheless, we have modified the sentence to soft the argument in the revised version (3rd paragraph, part 5).
paragraph 6: the arguments are interesting but the authors should not forget that, if domesticated starins are more often geterozygous than wild ones, it suggest that they not only more often originate from outbreeding but also include long distance introgressions. Unfortunately, long distance introgressions are not properly discussed in this review despite that fact that some of them have been shown to play an important role in the domestication. Refrences to such articles would be appropriate in paragraph 7. Similarly, loss of heterozygosity (LOH) is not mentionned in paragraphs 8 and 9 despite the fact that it is a major phenomenon in yeast strains (see for exemple the works of Magwene et al or Peter et al ).
>> We agree that introgressions play an important role in the domestication and niche adaptation of yeast, as we discuss in the manuscript using the Milk/Cheese lineage as an example. We have added more discussion on introgressions/HGT and related references in the revised version (2nd paragraph, part 7).
However, we have not found any evidence showing that introgressions contribute to the elevated heterozygosity of domesticated strains. We have detected as many introgression/HGT events in wild lineages as in domesticated lineages. The wild lineages from primeval forests harboring abundant introgressions/HGT are exclusively homozygous (Duan et al. 2018).
We agree that LOH may play a role in niche adaptation and diversification of domesticated strains and we regret that LOH was not discussed in the last version. We have added one more paragraph in part 7 for discussing studies on LOH (3rd paragraph, part 7). We think it is more appreciate to discuss LOH in part 7 than in parts 8 and 9.
paragraph 10. the conclusion may be expanded given above remarks.
>> One more paragraph (the 5th paragraph) for HGT/introgression has been added in part 10.
line 626: unique. Are the life cycle and mating system of S. cerevisiae so unique ? Actually, not if the authors would have paid little more attention on other Saccharomycotina species. Must be corrected.
>> We agree that ‘unique’ is not appropriately used here. The word has been deleted in the revised version.
Reviewer 2 Report
This is an interesting review of the eco-devo of S. cerevisiae mainly based on genomic information. The following revisions are recommended before publication. 1) L-95. The upper and lower parts of Fig. 1 are in a mirror image relationship. Does the authors need both parts? 2) L-190 The meaning of the color coding in Fig. 2 is not shown. What does the order of the horizontal axis mean? 3) L315. It is better to distinguish between wild and domestic yeast in Fig. 4? "Substitutions per site" is not shown in the figure. Is it possible to include the information of sequence diversity? 4) Please cite the following papers on the interaction between yeast and insects. https://doi.org/10.1371/journal.pone.0002873 5) The S. cerevisiae pan-genome is composed of 4,940 core ORFs and 2,856 ORFs that are variable within the population. Please discuss eco-devo from the perspective of the core gene and variable gene of pan-genome. 6) Can the authors make a comment on the number of out of China events?Author Response
Response to Reviewer 2 Comments
This is an interesting review of the eco-devo of S. cerevisiae mainly based on genomic information. The following revisions are recommended before publication.
>> We thank the generally positive comment on our manuscript. All the following specific comments and suggestions have been accommodated in the revised version of the manuscript.
1) L-95. The upper and lower parts of Fig. 1 are in a mirror image relationship. Does the authors need both parts?
>> In addition to aesthetics consideration, we use both parts to show that both MATa and MATα cells can switch mating type and the genotypes of the diploid cells formed the haplo-selfing of MATa and MATα cells may be different.
2) L-190 The meaning of the color coding in Fig. 2 is not shown. What does the order of the horizontal axis mean?
>> These have been explained in the revised legend of this figure.
3) L315. It is better to distinguish between wild and domestic yeast in Fig. 4? "Substitutions per site" is not shown in the figure. Is it possible to include the information of sequence diversity?
>> The wild and domesticated (liquid- and solid-state fermentation groups) are distinguished using branch lines with different colors as indicated on the upper left. The tree is a simplified schematic diagram and the branch lengths do not exactly correspond to the genetic distances between different lineages, thus “Substitutions per site" is not shown. The information has been added to the revised legend of Fig. 4. The sequence diversity of each lineage is shown in Fig. 5.
4) Please cite the following papers on the interaction between yeast and insects. https://doi.org/10.1371/journal.pone.0002873
>> Thanks for this suggestion. The paper (Coluccio et al., 2008) has been cited in related positions in the revised version of this manuscript.
5) The S. cerevisiae pan-genome is composed of 4,940 core ORFs and 2,856 ORFs that are variable within the population. Please discuss eco-devo from the perspective of the core gene and variable gene of pan-genome.
>> A short paragraph on the discussion of the core and variable genes has been added in part 10.
6) Can the authors make a comment on the number of out of China events?
>> It is an interesting question but the answer to this question depends on the answer to the single vs. multiple domestication events hypotheses debate. A comment has been added to the end of the second paragraph of part 10.